

# An intense 60-day weight-loss course leads to an 18 kg body weight reduction and metabolic reprogramming of soldiers with obesity

Exsal M. Albores-Méndez[1], Humberto Carrasco-Vargas[1], Samary Alaniz Monreal[2],
Rodolfo David Mayen Quinto[1], Ernesto Diderot López García[1], Gabriela Gutierrez Salmean[2], Karen Medina-Quero[1], Marco A. Vargas-Hernández[1], Cesar Vicente Ferreira Batista[1], Yamilé López-Hernández[3] and Robert Winkler[4]

[1] Escuela Militar de Graduados de Sanidad, Universidad del Ejército y Fuerza Aérea Mexicanos, Secretaría de la Defensa Nacional, Mexico City, Mexico
[2] Universidad Anáhuac, Huixquilucan, Mexico
[3] Laboratorio de Proteómica y Metabolómica de la Unidad de Ciencias Biológicas, Universidad Autónoma de Zacatecas, Zacatecas, Mexico
[4] Unidad de Genómica Avanzada, Cinvestav, Irapuato, Mexico

Corresponding author
Robert Winkler,
robert.winkler@cinvestav.mx

## ABSTRACT

Soldiers of the Mexican Army with obesity were subjected to an intense 60-day weight-loss course consisting of a controlled diet, daily physical training, and psychological sessions. The nutritional treatment followed the European Society of Cardiology (ESC) recommendations, incorporating elements of the traditional *milpa* diet in the nutritional intervention. The total energy intake was reduced by 200 kcal every 20 days, starting with 1,800 kcal and ending with 1,400 kcal daily. On average, the participants reduced their body weight by 18 kg. We employed an innovative approach to monitor the progress of the twelve soldiers who completed the entire program. We compared the untargeted metabolomics profiles of their urine samples, taken before and after the course. The data obtained through liquid chromatography and high-resolution mass spectrometry (LC-MS) provided insightful results. Classification models perfectly separated the profiles pre and post-course, indicating a significant reprogramming of the participants' metabolism. The changes were observed in the C1-, vitamin, amino acid, and energy metabolism pathways, primarily affecting the liver, biliary system, and mitochondria. This study not only demonstrates the potential of rapid weight loss and metabolic pathway modification but also introduces a non-invasive method for monitoring the metabolic state of individuals through urine mass spectrometry data.

## INTRODUCTION

Obese soldiers suffer similar health risks as civilians, such as hypertension, insulin resistance, and dyslipidemia, which can lead to chronic diseases and premature death (*Alberti et al.,*

*2009*). In addition, their reduced physical fitness and mobility can endanger them and their team during military operations. In recent years, there has been a growing concern about the prevalence of obesity among military personnel in the Mexican Army. In 2010, the obesity rate was 2.26% (*Gómez & Guzmán, 2019*). According to the Directive for the Prevention of Obesity in the Army and the Mexican Air Force, the prevalence of overweight and obesity increased to 4.52% in 2015 (*García-Chávez, Guzmán & Figueroa-Lara, 2021*). The rapid increase in obesity has led military leaders to implement weight loss programs for soldiers, aiming to improve their health. The army designed a 60-day weight loss course for rapid weight reduction. The program combines a calorie-restricted diet, exercise, and behavioral therapy.

For the study, obese personnel from the Mexican Army participated in the 60-day weight loss course at the Lifestyle Improvement and Health Center of the Physical and Aquatic Skills Program under the guidance of experienced researchers and healthcare professionals.

To achieve a weight reduction of 0.5–1.0 kg per week or 2.0–4.0 kg per month, the daily energy deficit must be 500–1,000 kcal alongside regular physical activity (*Bischoff & Schweinlin, 2020*). During the 60-day training program, the total energy intake is gradually reduced. In the first 20 days, 1,800 kcal are provided; in the next 20 days, 1,600 kcal; and in the last 20 days, 1,400 kcal. A further reduction with a food-based, low-calorie diet could lead to micronutrient deficiencies and is therefore not advisable (*Yumuk et al., 2015*).

To promote the participants' adherence to the diet, it should fit into its social and cultural context (*de Salud, 2012*; *Wharton et al., 2020*). In Mesoamerica, maize, beans, pumpkin, and other food plants are traditionally co-cultivated in a so-called '*milpa*' (*Lopez-Ridaura et al., 2021*). The *milpa* polyculture maintains biodiversity and ensures the self-sufficiency of small stakeholders. Thanks to the variety of the *milpa* diet, the nutritional requirements for macro- and micronutrients are covered (*Falkowski et al., 2019*; *Novotny et al., 2021*; *Méndez-Flores et al., 2023*; *Sánchez-Velázquez et al., 2023*). Proteins from locally available animals, such as fish and seafood, poultry, dairy products, and insects, complement the nutrition; however, red meat consumption is low (*Almaguer González et al., 2018*). The overall food composition of the *milpa* diet is similar to the Mediterranean diet, which is recommended for reducing the risk of cardiovascular diseases (*Visseren et al., 2021*). The consumption of junk food led to increased malnutrition, obesity, and related chronic diseases in the local population (*Leatherman & Goodman, 2005*). Thus, introducing snacks, *i.e.,* energy-dense, nutrient-poor, and salty food and sodas, into the training camp is prohibited.

The physical exercise program followed the recommendations of the Physical Activity Guidelines for Americans (*Piercy et al., 2018*). Since the training program is intense, only healthy individuals with no other diseases, such as diabetes, are admitted.

To investigate the impact of the 60-day weight-loss course on the participants' metabolism, we evaluated untargeted metabolomic profiles of twelve soldiers who successfully passed the complete program before and after the treatment.

## MATERIALS AND METHODS

### Participants and lifestyle improvement and health center

Every three months throughout the year, groups of military personnel with obesity who need to reduce their body weight to continue their activities within the Mexican Army are recruited at the Lifestyle Improvement and Health Center of the Physical and Aquatic Skills Program.

The Center aims to help soldiers regain their physical condition to fulfill their duties. During the 60-day training program, the participants cannot leave the Center and may only consume the food provided.

In the present study, 12 soldiers of the Mexican military with obesity who entered the Lifestyle Improvement and Health Center of the Physical and Aquatic Skills Program in Mexico City during the last recruitment period were included, all of whom signed an informed consent form and completed the program until the last day.

The Body-Mass Index (BMI) was calculated according to the World Health Organization (WHO) definition. Soldiers with a BMI equal to or higher than 25 were classified as overweight, and those with a BMI equal to or above 30 were classified as obese (*World Health Organization (WHO), 2021*).

All participants of the study were obese with a BMI >30 kg/m$^2$, with no upper limit, age, or sex restriction. However, since the training program is intense, only healthy individuals with no other diseases, such as diabetes, were admitted.

Clinical data were obtained according to Standard Operating Procedures (SOPs) in a military hospital's clinical laboratory. These methods include enzymatic, colorimetric, and photometric detection.

### Nutritional treatment

The nutrition followed the recommendations of the European Society of Cardiology (ESC) (*Visseren et al., 2021*). Still, it incorporated elements of the local *milpa* diet, such as regional vegetables and fruits, monounsaturated fatty acids provided by avocados, and whole grains from tortillas.

During the 60-day training program, total energy intake was gradually reduced by 200 kcal every 20 days under the guidance of a nutrition professional. Meals were prepared by cooks supervised by this same professional at the Lifestyle Improvement and Health Center facilities of the Physical and Aquatic Skills Program.

They started from 1,800 kcal and ended the last twenty days with a total consumption of 1,400 kcal per day. Macronutrient distribution over the 60-day period consisted of 50.1% to 53.41% carbohydrates, 18.26% to 20% proteins, and 28.33% to 30.6% lipids, with saturated fatty acids comprising 4.3% to 5.4%, polyunsaturated fatty acids 6.6% to 7.5%, monounsaturated fatty acids 13.08% to 15.8%, trans fatty acids ranging from 0% to 0.6%, and other fatty acids ranging from 2.3% to 3.71%. Analysis of diets was conducted using the Food Processor software, Version 11.11, ESHA Research (Salem, Oregon, United States). Details of the food composition are presented in Table 1.

The total energy intake was reduced by adjusting the portions' size and the food's caloric value. The dietary equivalents were prescribed according to the Mexican System

**Table 1  Energy content and macronutrient composition of the diet.** The nutrition was inspired by the local *milpa* diet, which is rich in vegetables and low in red meat. Using regional food reduces costs and supports the participant's adherence to the diet.

| Duration of diet (Days) | 0–20 | 21–40 | 41–60 |
|---|---|---|---|
| **Energy** (Kcal/day) | 1,800 | 1,600 | 1,400 |
| **Macronutrients** (% distribution) | | | |
| Carbohydrate | 53.41% | 51.1% | 50.1% |
| Protein | 18.26% | 20.0% | 19.2% |
| Fat | 28.33% | 28.9% | 30.6% |
|     Saturated | 4.47% | 4.3% | 5.4% |
|     Monounsaturated | 13.08% | 13.6% | 15.8% |
|     Polyunsaturated | 7.03% | 7.5% | 6.6% |
|     Trans fatty acid | 0.03% | 0.00% | 0.6% |
|     Others | 3.71% | 3.5% | 2.3% |
| **Compounds** | | | |
| Red meat (g per week) | 210 | 200 | 180 |
| Fish (servings) | 3 | 1–2 | 1-3 |
| Fish and poultry (servings) | 3 | 1–2 | 1–3 |
| Vegetables (g) | 450 | 500 | 600 |
| Fruits (g) | 650 | 500 | 400 |
| Nuts or seeds (servings) | 12–20 | 4–16 | 4–16 |
| Low fat dairy Products (servings) | 2 | 2 | 2 |
| Oil (cooking oil, olive oil, margarine) (servings) | 1–3 | 1–3 | 1–3 |
| Sodium (mg) | 1,489 | 1,262 | 1,147 |
| Dietary fiber (g) | 54.06 | 49.25 | 39.78 |

of Equivalent Foods (SMAE), with portion adjustments designed to gradually reduce 200 kilocalories every 20 days. Over the course of the dietary intervention, fruit equivalents were reduced from an initial average of 6.9 to 5.9 in the final 20 days. Concurrently, vegetable equivalents increased from 5.6 to 5.9, and cereal equivalents decreased from 9.3 to 6.5. Throughout the 60-day dietary protocol, dairy intake was consistently maintained at two equivalents per day. Fat intake, including those containing and devoid of protein, was systematically reduced from an average of 8.2 to 6.6 equivalents per day. Very low-fat food equivalents were decreased from 2.3 to one per day, while low-fat food equivalents were maintained between zero and one per day. Moderate-fat food equivalents were restricted to one per day, and moderate to high-fat food equivalents were limited to two per week.

During the intervention, the meal plan was divided into five sections, with ten menus for each 20 days. The diet was based on consuming vegetables such as nopal (cactus), mushrooms, and locally available fruits, including papaya, orange, and watermelon. Nuts or seeds such as peanuts or pistachios were included. Low-fat dairy products like yogurt, lactose-free milk, and panela cheese were also incorporated. Red meat intake was limited

to less than 200 g per week. Fish and poultry, including skinless chicken, egg, tuna, and locally sourced white fish, were consumed in portions ranging from one to three servings per day, while turkey ham was limited to less than two servings per week. Sugary drinks were avoided and replaced with unsweetened tea, hibiscus, or lemonade with sucralose. For food preparation, one to three servings of cooking, olive, or margarine were used in each meal.

## Physical activity program

The Physical Activity Guidelines for Americans suggest performing more than 150 to 300 min per week of moderate-intensity aerobic physical activity or more than 75 to 150 min per week of vigorous-intensity aerobic physical activity (*Piercy et al., 2018*). The purpose of the 60-day program was to progressively engage the participants in physical exercises to decrease their body mass index gradually.

The program was divided into nine weeks and distributed to progress the physical activity over 60 days. The timing and planned exercises were subject to and supervised at the sports instructor's discretion based on the physical capabilities of the participating military personnel in the study while still following the established program at the Lifestyle Improvement and Health Center of the Physical and Aquatic Skills Program.

A fixed schedule was followed throughout the seven days of the week to carry out the planned activities. Exercises were conducted three times daily from Monday to Saturday. On the seventh day, participants only engaged in 60 min of mild to moderate aerobic activities, with the remainder of the day allocated for rest and family activities at the improvement center. Joint lubrication and muscle activation exercises were performed at the beginning of each session.

Additionally, aerobic and resistance activities were included, gradually increasing in intensity from mild to vigorous. The intensity of these exercises was assessed using the Metabolic Equivalent of Task (MET) from the Compendium of Physical Activities 2011. Relaxation and respiratory recovery exercises were conducted at the end of each session. Detailed descriptions of the physical activity program by weeks and hours are provided in Table 2.

## Sample preparation

The urine samples were collected on the first day of admission to the Lifestyle Improvement and Health Center of the Physical and Aquatic Skills Program before starting the nutritional and physical activity intervention. The samples were obtained in the morning, as the first urine of the day and mid-stream, with participants in a fasting state and previously cleansed. Samples were collected in a sterile 120 ml container, with hands washed with soap and water previously to prevent contamination. The same procedure was repeated to collect the final urine sample, which was taken the morning after completing the 60-day intervention.

The urine samples were stored at −60 °C until processing. The processing of each urine sample was carried out following the URINE protocol (*Alseekh et al., 2021*). In a 1 ml microcentrifuge tube, 800 μl of cold methanol (MeOH) was added along with a 200 μl sample of urine from each participant. Similarly, a urine pool for each diet and a pool

**Table 2 Overview of the physical activity schedule.** The complete program takes nine weeks with increasing intensity.

| Week | 1 | 2 | 3 4 | 5 | 6 | 7 8 | 9 |
|---|---|---|---|---|---|---|---|
| **Activities from 06:00 to 07:00 hours** | Low to moderate intensity aerobic activities | Moderate to vigorous aerobic activities | Vigorous aerobic activities | Moderate to vigorous aerobic and resistance exercises | | | Vigorous aerobic and resistance exercises |
| From Monday to Saturday for 40 min | | | | | | | |
| Sunday for 60 min | | | Low to moderate intensity aerobic activities | | | | |
| **Activities from 11:30 to 13:30 hours** | | Low to moderate intensity aerobic and resistance activities | | Low to vigorous intensity aerobic and resistance activities | Moderate to vigorous intensity aerobic and resistance activities | Mainly vigorous intensity aerobic and resistance activities | Vigorous intensity aerobic and resistance activities |
| From Monday to Saturday for 120 to 240 min | | | | | | | |
| **Activities from 16:00 to 18:00 hours** | | | Walking or free sports | | | | |
| From Monday to Saturday for 120 min | | | | | | | |

for each time group were prepared. Ten quality control (QC) tubes containing similar proportions of all the samples were also included. Each tube was then agitated for 30 s, and the samples were incubated at $-20\,°C$ for 24 h, followed by centrifugation at 16,000 g for 15 min at four $°C$. After centrifugation, 200 µl of the supernatant was transferred to 1 ml tubes and evaporated to dryness using a Speed Vacuum Eppendorf concentrator at 10 $°C$. The dried extract was reconstituted in 200 µl of MeOH/water (1:1, v/v) and agitated for 1 min. Subsequently, the samples were centrifuged at 16,000 g for 15 min at 4 $°C$. The supernatant of each sample was transferred to UPLC-MS vials (Ultra High-Performance Liquid Chromatography with Mass Spectrometry) for subsequent injection. Three blanks containing the solvent used in sample processing (MeOH/H2O, 1:1, *v/v)* were included.

## High-performance liquid chromatography coupled to mass spectrometry

The samples were analyzed using a Waters XEVO G2-XS system with a quadrupole time-of-flight (QToF) mass analyzer. Modified methods described in 2022 (*Albores-Mendez et al., 2022*) were used to separate molecules. The Waters Corp (Milford, MA, USA) ACQUITY UPLC BEH C18, 130 Å, 1.7 µm, 2.1 mm × 100 mm BEH C18, 2.1 × 100 mm, 1.7 µm column was used. The mobile phases used solvent A, consisting of water with 0.1% formic acid, and solvent B, consisting of acetonitrile with 0.1% formic acid. All solvents used were LC-MS grade, acquired from Sigma Aldrich (St. Louis, MO, USA) and JT Baker (Phillipsburg, NJ, USA). The flow rate in the column was 0.1 mL/min until it was established at 0.4 mL/min, and the column temperature was maintained at 55 $°C$ during the process. The gradient of injection in the column started at 1% of solvent B and increased to 99% at minute 10, remaining at this injection gradient until minute 13. At minute 14, the injection gradient decreased to 1% and remained at that level until minute

17. The Waters XEVO G2-XS was used in high-resolution positive mode. A cone voltage of 40V and a capillary voltage of 3 kV were applied. The desolvation temperature was set at 450 °C, while the source temperature was 120 °C. The cone gas flow was set to 50 l/h, and the desolvation gas flow was set to 1000 L/h. A fragmentation energy ramp from 10 to 40 eV was applied, and the scan time was set at 0.2 s with a 30-second interval.

## Conversion of raw files to mzML

We used the ProteoWizard msconvert tool (https://proteowizard.sourceforge.io/) (*Kessner et al., 2008*) for converting the raw mass spectrometry data into the mzML community format.

## Mass spectrometry data pre-processing

For pre-processing the mass spectrometry data and the following statistical analyses, we used the web-based version of MetaboAnalyst (https://www.metaboanalyst.ca/) (*Xia et al., 2009*; *Pang et al., 2021*; *Wishart, 2020*). To improve subsequent statistical analyses, we applied data filtering (*Hackstadt & Hess, 2009*). We used the functions provided by MetaboAnalyst: an Interquartile range (IQR) filter of 40%, normalization on the median, square root transformation, and auto-scaling (mean-centered and divided by the standard deviation of each variable). The uploaded data set contained 24 samples. The pre-processing resulted in 1,377 features, and the data matrix was organized into two groups: before and after the treatment.

Figure 1 demonstrates the effect of the mass spectrometry data normalization. The normalized data were used for further statistical and pathway analyses.

### *Multivariate receiver operating characteristic curve-based exploratory analysis*

We applied an automated important feature identification and evaluated the performance of the models using three multivariate algorithms–support vector machines (SVM), partial least squares discriminant analysis (PLS-DA), and random forests. For classification and feature ranking, we applied the random forest algorithm. The receiver operating characteristic (ROC) curves were produced with a Monte-Carlo cross-validation (MCCV) method with balanced sub-sampling. For each calculation, two-thirds of the data were used to estimate the importance of the feature. The remaining third data portion was used for model validation. The individual model's performance and confidence could be calculated with multiple repetitions of this procedure.

## Global metabolomics and functional analysis

For global metabolomics and functional analysis, we defined a molecular weight tolerance of 10 ppm and used the Mummichog 1.0 (*Li et al., 2013*) and gene set enrichment analysis (GSEA) (*Subramanian et al., 2005*) algorithms with a *p*-value cutoff of 0.15. For the visual examination, we used a scatter plot, and for the peak annotation, the Homo sapiens (MFN) meta library, compiled from KEGG, BiGG, and the Edinburgh model (*Li et al., 2013*). We only considered pathways and metabolite sets with at least two entries and results with at least two hits.
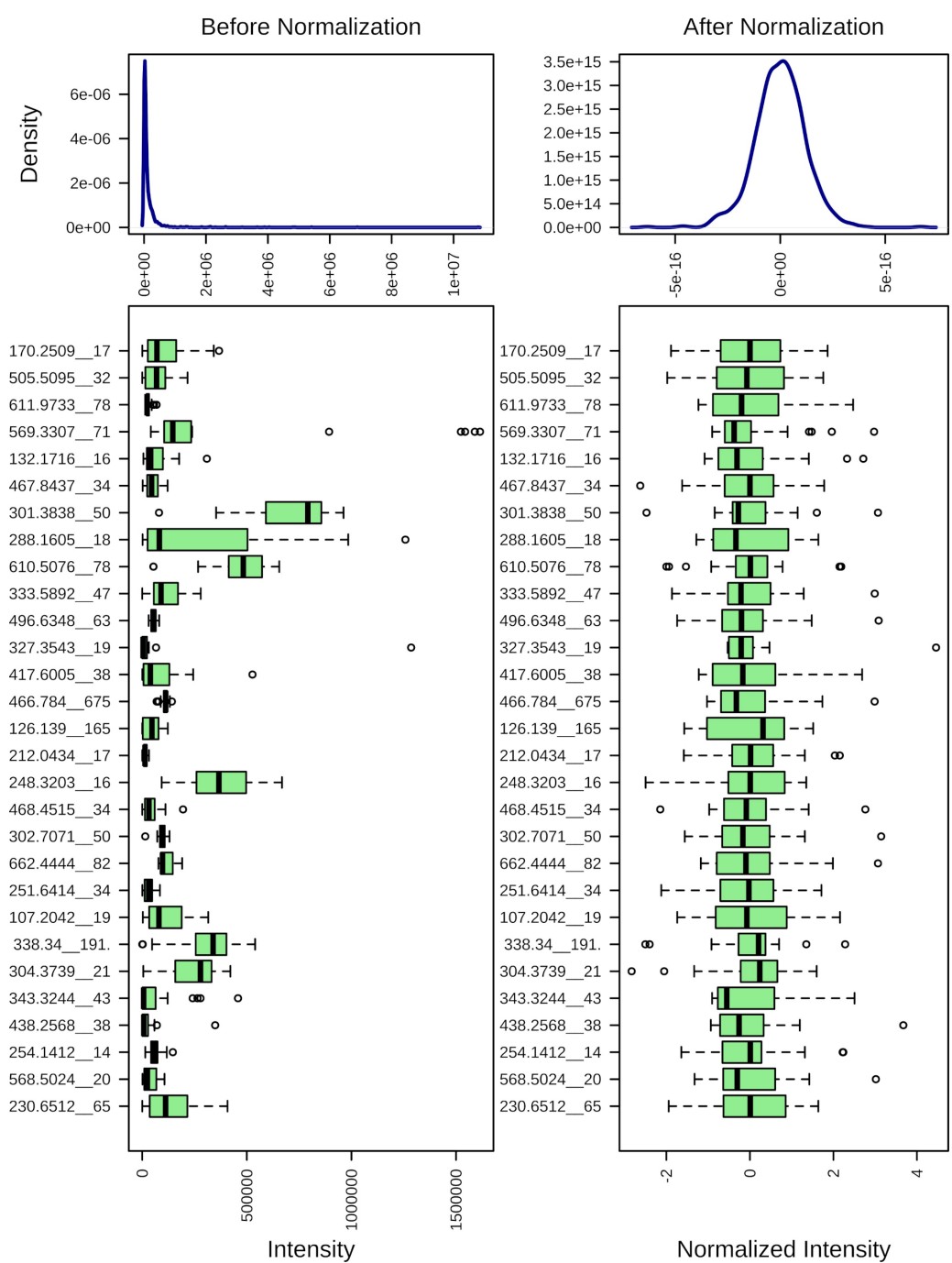

**Figure 1** **Normalization of the mass spectrometry data.**

## Institutional review board statement

This study complies with the International Ethical Guidelines for health-related research with human beings, elaborated by the Council for International Science Organizations Doctors (CIOMS) in collaboration with the World Health Organization (WHO). The
Research Committee and the Bioethics Committee of the Medical School approved this work. Escuela Militar de Medicina, Universidad del Ejército y Fuerza Aérea Mexicanos (reg. 0129012020.).

## RESULTS

### Clinical data of the participants

All twelve participants of the metabolic study, eleven men and one woman, were obese with a body mass index (BMI) above 30 kg/m$^2$. Notably, only soldiers without additional diseases were admitted to the course to ensure that they would resist the intense 60-day training program.

Table 3 summarizes the clinical data of the individuals who participated in the metabolic study before and after entering the 60-day training program.

### Weight reduction during the training program

Figure 2 shows the weight of the twelve soldiers before and after passing the training program. The mean weight decreased from 94.2 kg to 74.2 kg. Applying a pairwise $t$-test reported a $p$-value of $4.082 \cdot 10^{-9}$, with a mean difference of 18.075 kg for the participants' weight before and after the training program. The 95% confidence interval for the weight loss was between 15.668 kg and 20.482 kg.

The participants' average body mass index (BMI) was reduced from 32.8 kg/m$^2$ to 26.4 kg/m$^2$.

Consequently, the protocol is highly efficient for significantly reducing the participants' body weight.

### Classification by metabolic profiles

To test if the treatment impacts the metabolic profile of the participants, we subjected the data of all twelve datasets before and after the 60-day training to a random forest algorithm and evaluated the generated models.

The metabolic datasets before and after the treatment could be distinguished without error by the Random Forest model, indicating distinct metabolic profiles of the individuals before and after the treatment.

Figure 3 presents the classification of the sample groups according to their metabolic profiles. The predictive Random Forest model permits reliable discrimination between samples before and after the treatment. Figure 3A shows the correct clustering of the samples. The graphs B and C of the exact figure indicate that as few as five features of the metabolic profile are sufficient for identifying a sample with an error of about 1%. Figure 3D lists the important variables for one model.

The highly reliable predictive model demonstrates substantial differences in the metabolic state of the soldiers before and after the training program. In addition, suitable sets of a few (five to ten) compounds could be used for monitoring the training state of participants of a weight loss program.

**Table 3   Clinical data of the metabolic study participants before and after the 60-day training program.**
Only individuals without further diseases were admitted. SD–Standard deviation.

| Parameter | Before | | | After | | |
|---|---|---|---|---|---|---|
| | Range | Mean | SD | Range | Mean | SD |
| Weight (kg) | 77.0–107.4 | 92.2 | 8.6 | 64.2–85.5 | 74.2 | 7.9 |
| Height (m) | 1.57–1.8 | 1.68 | 0.07 | 1.57–1.8 | 1.68 | 0.07 |
| Body-Mass-Index (BMI) (kg/m2) | 30.1–39.0 | 32.8 | 3.2 | 22.7–32.7 | 26.4 | 2.7 |
| Sex | | | | | | |
| Male | 11 (98%) | | | 11 (98%) | | |
| Female | 1 (8%) | | | 1 (8%) | | |
| Age (y) | 21–37 | 30.9 | 5.8 | 21–37 | 30.9 | 5.8 |
| Glucose (mg/dL) | 75–98 | 83.5 | 6.8 | 73–95 | 86.8 | 6.9 |
| Urea (mg/dL) | 21.4–32.1 | 27.4 | 3.9 | 12.8–34.2 | 23.9 | 6.2 |
| Blood urinary nitrogen (BUN) (mg/dL) | 10–15 | 12.8 | 1.8 | 6–16 | 11.2 | 2.9 |
| Serum creatinine (mg/dL) | 0.7–1.2 | 0.9 | 0.1 | 0.7–1.1 | 0.9 | 0.1 |
| Cholesterol total (mg/dL) | 112–199 | 157.9 | 25.6 | 105–95 | 169.5 | 24.2 |
| Triglycerides (mg/dL) | 49–220 | 95.2 | 46.6 | 58–184 | 108 | 35.1 |
| HDL cholesterol (mg/dL) | 27–57 | 42.6 | 10 | 30–65 | 46.6 | 13.5 |

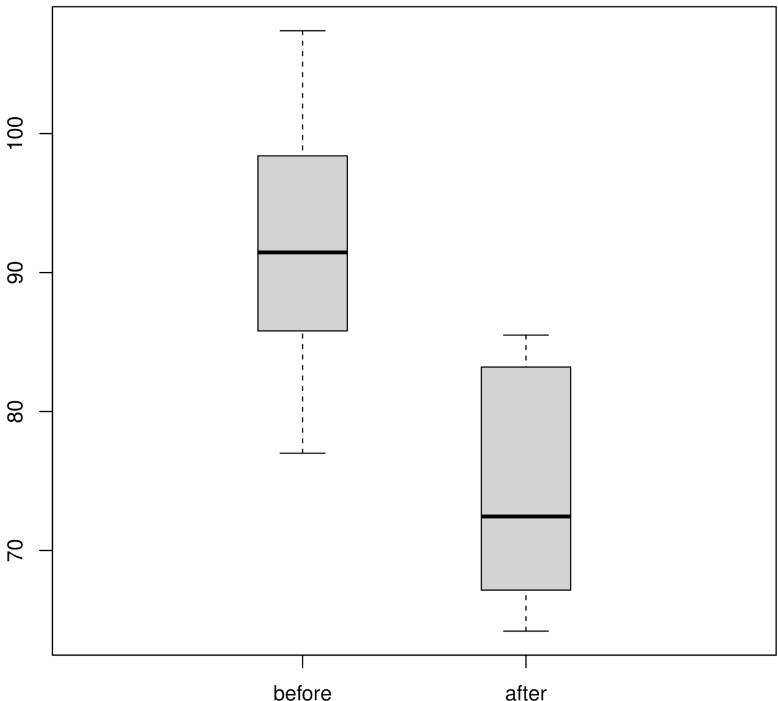

**Figure 2   Weight of the participants before and after the 60-days training program.**

## Metabolic pathways

Table 4 lists the most prominent differential metabolic pathways identified by the combined Mummichog (*Li et al., 2013*) and GSEA (*Subramanian et al., 2005*) analyses.

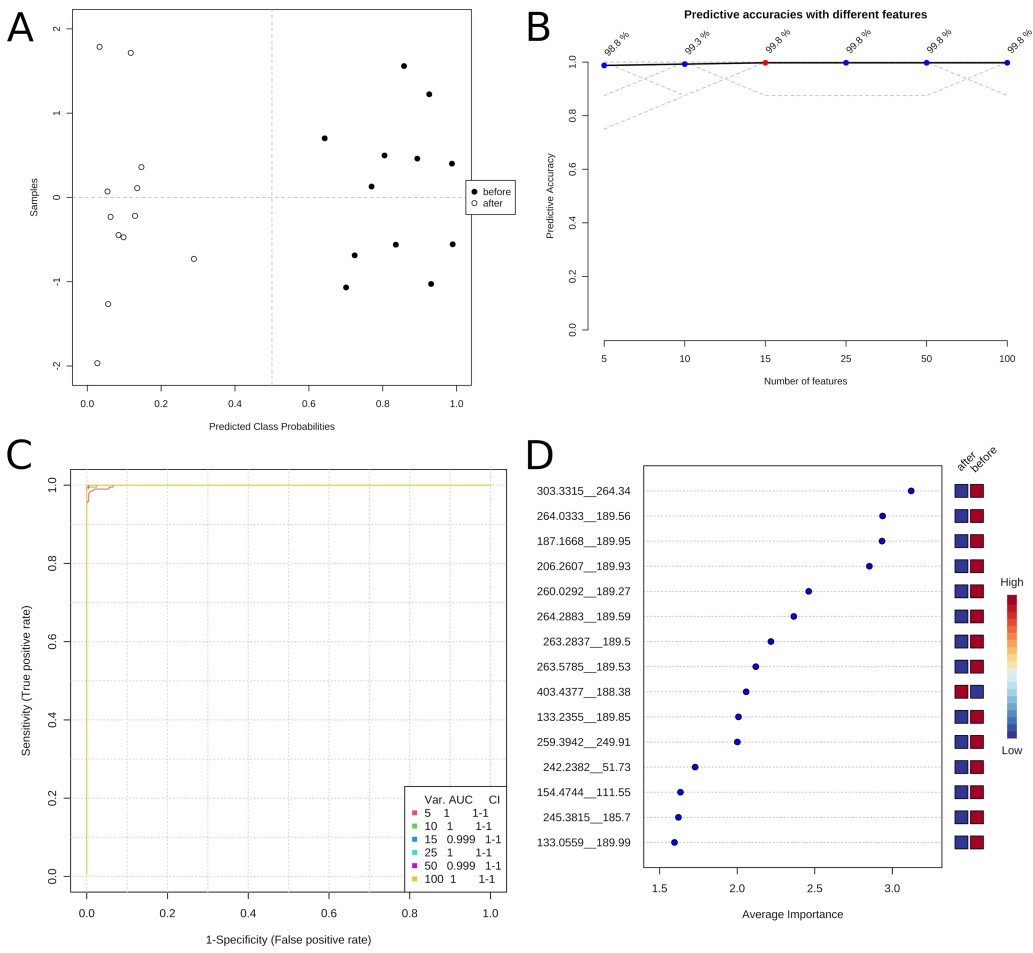

**Figure 3** Classification of the metabolomic profiles before and after treatment with predictive Data Mining. (A) Predicted class probabilities. (B) Predictive accuracies with different numbers of features. (C) Reciever operating characteristic (ROC) curves. (D) Variable importance analysis.

As expected, pathways related to fat, sugar, and nutrition-related metabolism, such as saturated fatty acids beta-oxidation, bile acid biosynthesis, purine metabolism, and the TCA cycle, were modified.

Most differential features can be annotated to amino acid metabolism, including the routes of $\beta$-alanine, tryptophan, methionine and cysteine, lysine, tyrosine, and related pathways, such as urea/amino group metabolism.

Another large group is related to the metabolism of vitamins, such as C, D3, E, C21 steroid hormones, and C1 metabolism.

In addition, the caffeine and porphyrin metabolism were affected.

## DISCUSSION

The nutritional intervention was based on the European Society of Cardiology (ESC) in 2021, which advocates for a predominantly plant-based dietary pattern, such as the

**Table 4  Integrated pathway analysis with Mummichog and GSEA, ordered by combined *p*-values.**

| Pathway | Size | Hits | Mummichog Pvals | GSEA Pvals | Combined Pvals |
|---|---|---|---|---|---|
| Biopterin metabolism | 22 | 2 | 0.1685 | 0.04167 | 0.04183 |
| Saturated fatty acids beta-oxidation | 36 | 6 | 0.4268 | 0.01852 | 0.04617 |
| Vitamin D3 (cholecalciferol) metabolism | 16 | 6 | 0.4268 | 0.01852 | 0.04617 |
| Caffeine metabolism | 11 | 6 | 0.4268 | 0.02083 | 0.05088 |
| Bile acid biosynthesis | 82 | 19 | 0.834 | 0.01613 | 0.07141 |
| Ascorbate (Vitamin C) and Aldarate | 29 | 7 | 0.4781 | 0.04 | 0.09479 |
| Beta-Alanine metabolism | 20 | 3 | 0.242 | 0.1698 | 0.1723 |
| Porphyrin metabolism | 43 | 6 | 0.4268 | 0.1111 | 0.192 |
| Tryptophan metabolism | 94 | 25 | 0.9078 | 0.05714 | 0.2054 |
| Vitamin E metabolism | 54 | 7 | 0.4781 | 0.1154 | 0.215 |
| TCA cycle | 31 | 5 | 0.3707 | 0.1887 | 0.256 |
| Vitamin B3 (nicotinate and nicotinamide) | 28 | 5 | 0.3707 | 0.3585 | 0.4011 |
| Purine metabolism | 80 | 6 | 0.4268 | 0.3125 | 0.4021 |
| Glutathione Metabolism | 19 | 4 | 0.3093 | 0.617 | 0.5069 |
| Methionine and cysteine metabolism | 94 | 8 | 0.5249 | 0.48 | 0.5993 |
| Urea cycle/amino group metabolism | 85 | 12 | 0.6746 | 0.5238 | 0.721 |
| Aspartate and asparagine metabolism | 114 | 21 | 0.8634 | 0.4103 | 0.7219 |
| C21-steroid hormone biosynth. and met. | 112 | 22 | 0.8761 | 0.4872 | 0.7902 |
| Lysine metabolism | 52 | 12 | 0.6746 | 0.6905 | 0.8217 |
| Tyrosine metabolism | 160 | 27 | 0.9244 | 0.6757 | 0.9186 |

DASH (Dietary Approaches to Stop Hypertension) and the Mediterranean diet. The DASH pattern suggests that 27% of calories should come from total fat, 18% from proteins, and 55% from carbohydrates, with an emphasis on consuming at least 30 g of fiber daily and limiting sodium intake to 2,300 mg per day (*Eckel et al., 2014*; *Campbell, 2017*; *Visseren et al., 2021*).

The ESC emphasizes the importance of consuming less than 10% of total energy from saturated fats and reducing red meat consumption to a maximum of 350-500 g per week. Recommendations include consuming fish 1 to 2 times per week, 30 g of unsalted nuts daily, over 200 g of fruits daily, 30 to 45 g of fiber daily, and over 200 g of vegetables per day (*Visseren et al., 2021*). The source and quality of consumed products are essential, as emphasized in the Mediterranean diet, which provides vital macro and micronutrients such as monounsaturated fatty acids from olive oil, vitamins and minerals from fruits and vegetables, proteins and fatty acids from nuts and almonds, and fiber from whole grains such as oats (*Figueroa et al., 2021*).

The *milpa* diet of the 60-day weight-loss program closely follows the DASH (*Campbell, 2017*) and ESC (*Visseren et al., 2021*) recommendations and, therefore, can be considered suitable for the effective prevention of cardiovascular diseases. In addition, it respects traditional food habits and is available from local producers at high quality and low cost. Consequently, it is more likely that the cured patients will consume healthy food after
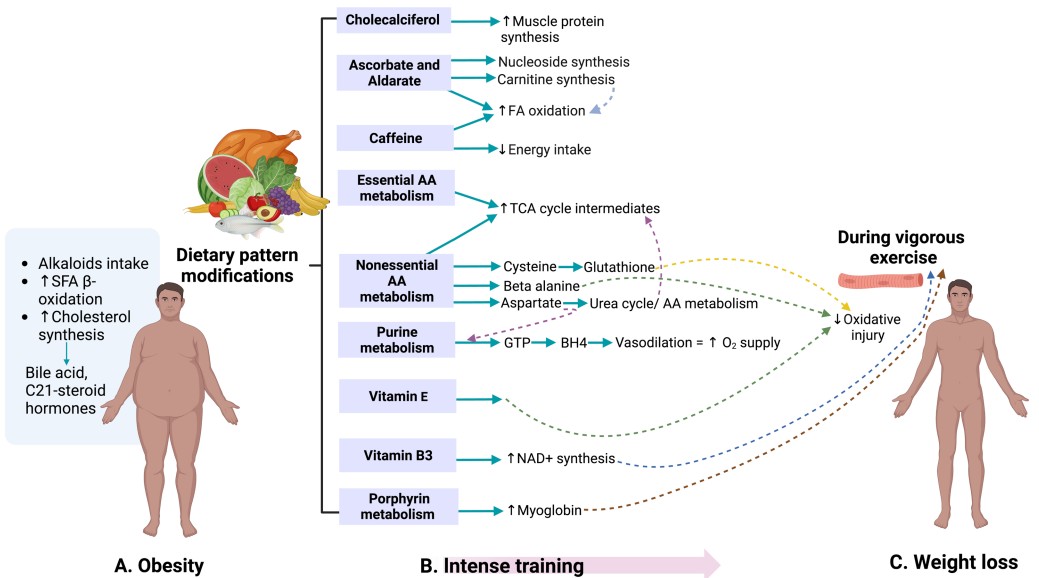

**Figure 4** The combination of the *milpa* diet and an intense training program lead to a significant weight reduction and a modification of metabolic profiles. Created with BioRender.com.

successfully finishing the program and maintain their improved physical state (*de Salud, 2012*; *Wharton et al., 2020*).

Figure 4 summarizes the overall effects of the 60-day weight-loss program. The controlled nutrition with defined composition and reduced energy uptake, and the intense physical training cause synergistic changes because muscles are built up, aerobic fitness is increased, and body fat is consumed for energy production (*Tremblay, Simoneau & Bouchard, 1994*; *Maillard, Pereira & Boisseau, 2018*).

Through the pathways identified in the study of urinary metabolome, the presence of metabolites associated with dietary sources attributable primarily to the development of obesity was observed. Metabolites from alkaloid biosynthesis, such as caffeine found in sugary drinks (*Reyes & Cornelis, 2018*), were consumed by 76.3% of the adult population in Mexico in 2022 (*Gaona-Pineda et al., 2023*). During the 60-day weight-loss program, no sources of caffeine, such as coffee, tea, or caffeine-containing sodas, were supplied. Thus, the observed differences can be attributed to detoxification.

Metabolites from saturated fatty acid oxidation pathways are found in dairy, butter, processed red meats, desserts, and cereals (*Steur et al., 2021*).

Similarly, products from bile acid synthesis and C-21 steroid hormones, compounds derived from cholesterol (*Li & Chiang, 2009*; *Strushkevich et al., 2011*), have been observed to increase in production with the presence of obesity (*Auley, 2020*). These pathways can be observed in Fig. 4A. These results are consistent with a urinary metabolomics study conducted in Saudi Arabia in 2016 among participants aged 18 to 40, where bile acids and carnitine were positively associated with BMI (*Ahmad et al., 2017*).

The lifestyle changes of the participants during military training were reflected in both the soldiers' weight and the metabolites, as shown in Fig. 4B. Compounds of vitamin D3 metabolism are likely to be related to the improvement in participants' physical performance and the increase in muscle protein production (*Carswell et al., 2018*). The presence of vitamin C and alderate, associated with nucleoside synthesis and carnitine (*Steenbergen et al., 2018*), were detected. The latter is essential in fatty acid oxidation (*Johnston, 2005*). Caffeine has been related to increased fatty acid oxidation and, in turn, decreased caloric intake (*Tabrizi et al., 2019*). The present study's dietary program was caffeine-free. However, coffee, tea, or other sugar-free drinks containing caffeine could benefit future diets.

The metabolism of essential amino acids such as lysine, methionine, and tryptophan leads to the formation of intermediates in the tricarboxylic acid cycle. It contributes to muscular activity during physical training (*Council, 1989*). Lysine produces acetyl-CoA, while methionine generates succinyl-CoA and is fundamental in synthesizing cysteine, a non-essential amino acid. Tryptophan is involved in acetyl-CoA, $NAD^+$, and NADP metabolism and participates in serotonin synthesis along with tetrahydrobiopterin (*Longo, 2009*; *Lieberman & MD, 2017*).

In a past study, we classified normal-weight, overweight, and military personnel with obesity by metabolic profiles. We discovered similar metabolites as biomarkers, such as those related to tryptophan metabolism, tyrosine, and the urea cycle, which converged in the pathway of S-adenosyl-L-methionine (SAM) (*Albores-Mendez et al., 2022*). This molecule is essential in methyl group donation, and its importance was observed in a 2007 study with a diabetic rat model, where it was significantly associated with increased mitochondrial DNA density in skeletal muscle and a notable improvement in insulin sensitivity (*Jin et al., 2007*). Insulin is crucial for blood flow in skeletal muscle, which becomes especially relevant during exercise (*Mahmoud et al., 2015*).

Amino acids such as tyrosine, which converts to fumarate, and aspartate, which gives rise to asparagine from oxaloacetate, were found. Aspartate is related to the urea cycle and amino group metabolism, culminating in fumarate synthesis and participating again in the Krebs cycle (*Lieberman & MD, 2017*). Additionally, aspartate acts as a precursor in the purine pathway, leading to the production of guanosine triphosphate, which is necessary for tetrahydrobiopterin synthesis (*Thöny, Auerbach & Blau, 2000*). The latter is essential in nitric oxide production, leading to vasodilation and increased muscle oxygen supply during exercise (*Mahmoud et al., 2015*).

Metabolic pathways of other non-essential amino acids, such as cysteine and beta-alanine, which are required in synthesizing compounds essential for protection against oxidative injury in the muscles during vigorous exercise, were identified. These pathways lead to glutathione production from cysteine and carnosine from beta-alanine (*Simioni et al., 2018*). During training, vitamin E also plays a protective role against damage caused by oxidative stress (*Meydani et al., 1993*). Vitamin B3 (nicotinate and nicotinamide) is essential in synthesizing $NAD^+$ molecules, whose levels increase during physical activity and act as antioxidants (*Ji & Yeo, 2022*). These pathways contribute to weight loss, as shown in Fig. 4C.
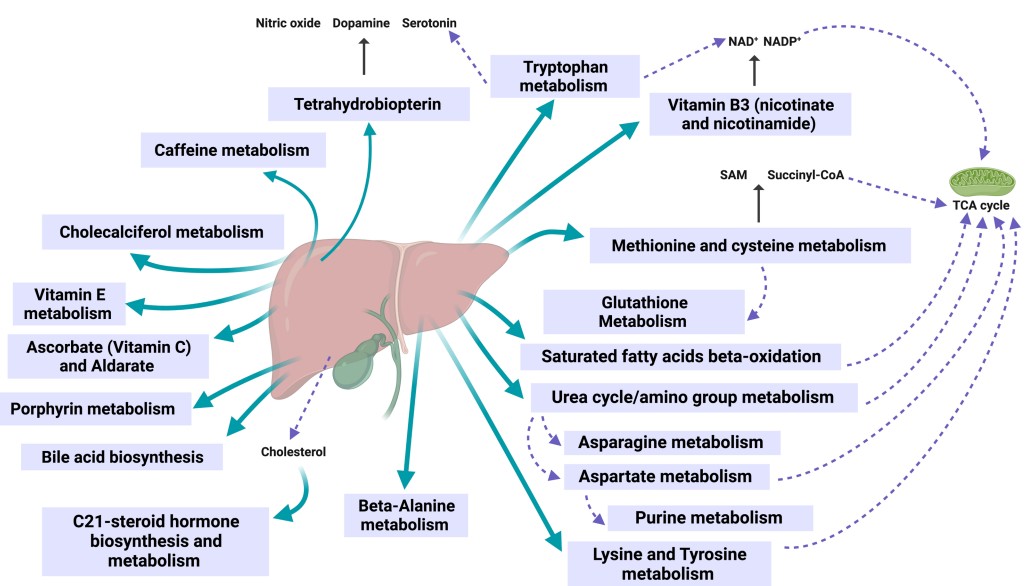

**Figure 5** Observed metabolic changes during the 60-day weight loss program are mainly localized in the liver, bile, and mitochondria. Created with BioRender.com.

Figure 5 illustrates how the obtained pathways converge in the liver and bile ducts, either in their synthesis, metabolism, or excretion.

The liver and the bile play a central role in lipid digestion, storage, and reuse (*Nguyen et al., 2008*; *Rui, 2014*). In addition, the liver participates in the metabolism of (especially liposoluble) vitamins (*Vranić, Mikolašević & Milić, 2019*; *Tanumihardjo, 2021*) and is vital for detoxification (*Grant, 1991*) (see Fig. 4). Obesity increases the risk of developing fatty liver disease (steatosis) (*Milić, Lulić & Štimac, 2014*). Such nonalcoholic fatty liver disease (NAFLD) can further progress to nonalcoholic steatohepatitis (NASH), liver cirrhosis, and cancer, with fatal consequences for the patient (*Dietrich & Hellerbrand, 2014*; *Lu et al., 2018*). In the stage of NAFLD, the liver can still regenerate by appropriate changes to the lifestyle and diet (*Allaire & Gilgenkrantz, 2020*). Our results indicate that the 60-day program activates essential liver functions and might contribute to its recovery.

## CONCLUSIONS

The 60-day weight-loss program, consisting of a strict diet plan based on a traditional *milpa* diet and intense physical activity, significantly reduced the body weight of the soldiers with obesity (by 18 kg on average) and reprogrammed their metabolic pathways. The program helps the participants to meet their job's physical requirements, to improve their overall health, and to stop the progression of obesity to develop a severe metabolic syndrome.

The twelve participants were between 21 and 37 years old and had no other health issues. Thus, correcting the body weight of persons with obesity before secondary diseases is strongly recommended for preventing later, possibly life-threatening, complications.

Metabolic pathway reconstruction from untargeted LC-MS data of urine suggested the modulation of the C1-, vitamin, amino acid, and energy metabolism, thus indicating the central role of the liver, the biliary system, and mitochondria in the physiological processes of the diet. Measuring related biomarkers could support the monitoring of weight-loss programs in future studies.

## ACKNOWLEDGEMENTS

We thank the Military Graduate School of Health (E.MG.S.) for providing the facilities for storing and processing the samples, and the CINVESTAV Irapuato for providing the facilities for the analysis and writing.

### Funding

The Budget Program A022 financially supported the project Military Research and Development in Coordination with Public Universities, Public Higher Education Institutions, and other Public Research Centers. Secretary of National Defense, Mexico. The Research and Development Center of the Mexican Army and Air Force (CIDEFAM) also provided financial support. The funders had no role in study design, data collection and analysis, decision to publish, or preparation of the manuscript.

### Grant Disclosures

The following grant information was disclosed by the authors:
The project Military Research and Development in Coordination with Public Universities, Public Higher Education Institutions.
Secretary of National Defense, Mexico.
The Research and Development Center of the Mexican Army and Air Force (CIDEFAM).

### Competing Interests

Robert Winkler is an Academic Editor of PeerJ and a Section Editor of PeerJ Plant Biology.

### Author Contributions

- Exsal M. Albores-Méndez conceived and designed the experiments, performed the experiments, analyzed the data, prepared figures and/or tables, authored or reviewed drafts of the article, and approved the final draft.
- Humberto Carrasco-Vargas analyzed the data, authored or reviewed drafts of the article, and approved the final draft.
- Samary Alaniz Monreal analyzed the data, authored or reviewed drafts of the article, and approved the final draft.
- Rodolfo David Mayen Quinto performed the experiments, authored or reviewed drafts of the article, and approved the final draft.
- Ernesto Diderot López García analyzed the data, authored or reviewed drafts of the article, and approved the final draft.

- Gabriela Gutierrez Salmean analyzed the data, authored or reviewed drafts of the article, and approved the final draft.
- Karen Medina-Quero performed the experiments, authored or reviewed drafts of the article, and approved the final draft.
- Marco A. Vargas-Hernández performed the experiments, authored or reviewed drafts of the article, and approved the final draft.
- Cesar Vicente Ferreira Batista performed the experiments, analyzed the data, authored or reviewed drafts of the article, and approved the final draft.
- Yamilé López-Hernández conceived and designed the experiments, performed the experiments, analyzed the data, authored or reviewed drafts of the article, and approved the final draft.
- Robert Winkler conceived and designed the experiments, analyzed the data, prepared figures and/or tables, authored or reviewed drafts of the article, and approved the final draft.

## Human Ethics

The following information was supplied relating to ethical approvals (i.e., approving body and any reference numbers):

The Research Committee and the Bioethics Committee of the Medical School approved this work. Escuela Militar de Medicina, Universidad del Ejército y Fuerza Aérea Mexicanos (reg. 0129012020.).

## Data Availability

The mass spectrometry data and the sample descriptions at Zenodo: Winkler, R., & Albores-Méndez, E. M. (2023). Metabolic profiles of obese soldiers before and after passing an intense 60-days weight-loss course (SEDENA) [Data set]. Zenodo. https://doi.org/10.5281/zenodo.8240190. The dataset was uploaded directly to the MetaboAnalyst web platform https://metaboanalyst.ca (*Wishart, 2020*; *Xia et al., 2009*; *Pang et al., 2021*) for pre-processing and statistical analyses.

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
