# Peer review of "An intense 60-day weight-loss course leads to an 18 kg body weight reduction and metabolic reprogramming of soldiers with obesity"

_PeerJ, doi:10.7717/peerj.17757_

## Round 0.1 · original submission · Minor Revisions

Thank you for submitting your manuscript, "An Intense 60-Day Weight-Loss Course: Impact on Body Weight and Metabolic Changes in Obese Soldiers," to our journal. Two reviewers have evaluated your manuscript, and we have identified some areas where additional information would enhance the clarity and comprehensiveness of your study.

Please address the raised questions for clarification. These adjustments will ensure the manuscript meets our publication standards.

Kind regards,

·

Basic reporting

The Manuscript “An intense 60-day weight-loss course leads to an 18 kg body weight reduction and metabolic reprogramming of soldiers with obesity” describes a small study that put soldiers with obesity in a 60-day weight loss intervention where energy intake was reduced to 1400 kcal/day by the end of the intervention. Untargeted metabolomics was used to detect changes pre-post intervention. It is no surprise that this highly intense program was effective for reducing body weight; however, the novelty of this pilot study was providing evidence for specific metabolic adaptations induced by rapid weight loss. This is a well-designed study, although it is not clear if this was a controlled feeding study, a live-in study, or (if not) if they had any way of tracking compliance to the dietary intervention.

Abstract:
It states that energy intake was “gradually reduced to 200 kcal every 20 days”. That seems to say that every 20 days, EI was 200 kcal. Guessing this should read “EI was reduced by 200 kcal every 20 days…”.

Introduction:
Opening paragraph does not convince reader obesity in the military is a major problem. Has the obesity rate changed since 2015? 4.5% seems pretty low. Wondering if we even need to bring obesity and the military into this… stating that rapid weight loss is often desired and a better understanding of the metabolic implications are needed may be a better angle to take here- especially since this should be framed as more of a mechanistic, early-phase study and not one to establish a weight loss program with only 12 participants.
Second paragraph, line 48, states that for a weight reduction of 0.5-1.0 kg/wk, daily energy intake must be reduced by 500-1,000 kcal. This is simply not true, it is very possible to lose this much weight without reducing energy intake by 500-1000 kcal and will vary greatly between individuals (400 lb male vs 200 lb female for example). The reference used here states that a 0.5-1.0 kg/week weight loss is realistic and that an energy deficit of at least 500 kcal is needed, which is not the same as reducing energy intake by 500 kcal. So this needs to be changed to come off the “must” or revise to use the idea of energy deficit as opposed to just reductions in energy intake.
Line 66: change “high-caloric” to “energy dense”.
Paragraph starting at line 68: seemed to introduce the weight loss program (first sentence), and the goes on to focus on the physical activity aspect. But this was done after the paragraphs speaking to the dietary aspect, so this threw me off a bit. Probably need to come out and say this first sentence before the second paragraph, clarifying that the program included both the exercise component and dietary restriction component. Then include details on both the dietary and physical activity components.

Experimental design

Materials and Methods:
Line 88 states the BMI requirements and defined BMI, which is perfect (guessing the “$” was a mistake though). Lines 89-93 go on to detail how BMI is calculated and defined, which is definitely not needed since you already defined BMI, a very well-known metric. Similarly, lines 231-232 of the results states that all participants were obese, which is not needed as we know that this was the requirement.
Line 104: paragraph shouldn’t start with the word “they”, not sure who “they” is.
Not entirely clear if food was provided (controlled feeding) or if a meal plan was just given and assumed to be followed?
Lines 122-128 seemed to be a repeat of lines 70-76, except a different reference was used. I would suggest revising the first mention of the PA program as noted in prior comment.
Line 152: is there a reason for saying “after completing the 20-day phase under 1400 kcal daily” and not “after completing the 60 day intervention”? it would seem the later is easier to understand.

Validity of the findings

Discussion:
Not sure the first 2 paragraphs are best for the discussion, seems to be describing the dietary intervention as in the methods.
In discussing the metabolomics, it was stated that caffeine metabolism was effected. This was brought up in line 303, giving rise to the idea that these metabolites were higher at baseline because of sugary drink consumption. Line 318 seems to hint that caffeine metabolism was increased post intervention as this is related to fatty acid oxidation and energy intake- although caffeine intake increasing FAO and reducing energy intake is not the same as different metabolites related to caffeine metabolism found in urine after weight loss. A good description of the other metabolites is provided though.
Conclusions:
Conclusions should be more focused on the specific metabolic findings. It is obvious such a program will promote weight loss, but this study was able to provide novel metabolomics data. Conclusion here should be focused on what this metabolomics data means and what we can use it for in the future.

·

Basic reporting

The introduction provides a clear overview of the issue, i.e., obesity among soldiers, its health implications, and the need for interventions. It effectively sets the context for the study. The mention of micronutrient deficiencies with further calorie reduction is relevant and emphasizes the need for a balanced diet during weight loss programs.

Some points should be corrected and clarified. The details as below,

Clarify how the macronutrient distribution was determined. Was it based on individual dietary needs or general guidelines?

Line 94 The clinical data were determined using standard methods. Method should be cited here by another study please.

Analysis of diets was conducted using the Food Processor software. What is the software name, model number company, manufacturer city and country.

Make sure all the units, formulas and abbreviations used in the manuscript are according to the journal’s guidelines, for this please check the author guideline section of the journal.

There is some typo and grammatical mistake should be rectifying in the revised version of the manuscript.

The authors should double check the citation format in main body of manuscript.

The study is novel, engaging, and worth to be improved in its structural form.

Experimental design

Line 84 Clarify the term "improvement center" and provide more context about its role in recruiting military personnel.

Specify the criteria used for selecting participants with obesity. For example, mention if there were any exclusion criteria apart from having a BMI >30 kg/m².

Provide more details about the gradual reduction of total energy intake. How was this reduction achieved in terms of specific food items or portion sizes?

Clarify how the macronutrient distribution was determined. Was it based on individual dietary needs or general guidelines?

Explain the role behind using multivariate algorithms for exploratory analysis and how they contribute to the study's objectives.

If manuscript is in English then authors should submit an English version of ethical statement.

Validity of the findings

Although the study is good in terms of its purpose but manuscript is lacking specifically about its structure.
In conclusion you did not mention how much weight loss you achieved in soldiers? Conclusions should be linked with the goals of the study.

Additional comments

Here are some suggestions about long sentences into short and readable form.
BMI formula should be mention in correct form.
Line 36-38
Original: Soldiers with obesity suffer the same health risks as the civilian population, such as hypertension, insulin resistance, and dyslipidemia, eventually leading to chronic diseases and premature death (Alberti et al., 2009).
Revised: Obese soldiers face similar health risks as civilians, such as hypertension, insulin resistance, and dyslipidemia, which can lead to chronic diseases and premature death (Alberti et al., 2009).

Somewhere authors used milpa diet as italicized and somewhere don’t…make it uniform throughout the document.

Line 38-39 Original: In addition, their reduced physical fitness and mobility can get them and their team into dangerous situations during military operations.
Revised: Their reduced physical fitness and mobility can endanger them and their team during military operations.

Line 43-45 Original: The fast-rising obesity rate has prompted military leaders to take action to address the issue and implement programs to help soldiers lose weight and improve their overall health.
Revised: The rapid increase in obesity has led military leaders to implement weight loss programs for soldiers, aiming to improve their health.

Line 45-46 Original: The army developed a 60-day weight loss course, allowing participants to lose weight quickly.
Revised: The army designed a 60-day weight loss course for rapid weight reduction.

Line 48-50 Original: For a weight reduction of 0.5 to 1.0 kg per week, or 2.0 to 4.0 kg per month, the daily energy intake must be reduced by 500 to 1000 kcal, combined with regular physical activity (Bischof and Schweinlin, 2020).
Revised: To achieve weight reduction goals (0.5-1.0 kg per week or 2.0-4.0 kg per month), daily energy intake should be reduced by 500-1000 kcal alongside regular physical activity (Bischof and Schweinlin, 2020).

Line 64 The consumption of Junk Food…. Why it is italic here?
Line 68-70 Original: The study participants, personnel from the Mexican Army who were overweight or obese, underwent physical activation for 60 days at the Lifestyle Improvement and Health Center of the Physical and Aquatic Skills Program.
Revised: Overweight or obese personnel from the Mexican Army participated in a 60-day physical activation program at the Lifestyle Improvement and Health Center.

Line 218 World Organization for Health (WHO)…kindly correct it please.

---

## Round 0.2 · accepted · Accept

Dear Authors,

Congratulations on the acceptance of your paper! I am pleased to confirm that you have successfully addressed all of the reviewers' comments. Thank you for your diligence and contribution to our journal.

Best regards,

·

Basic reporting

The authors did a good job at responding to my original comments. There may be a few minor typographical errors, but otherwise this is a quality publication.

Experimental design

The authors did a good job at responding to my original comments. There may be a few minor typographical errors, but otherwise this is a quality publication.

Validity of the findings

The authors did a good job at responding to my original comments. There may be a few minor typographical errors, but otherwise this is a quality publication.

Additional comments

The authors did a good job at responding to my original comments. There may be a few minor typographical errors, but otherwise this is a quality publication.

·

Basic reporting

Dear Authors,
Thank you for submitting the revised version of your manuscript titled "An intense 60-day weight-loss course leads to an 18kg body weight reduction and metabolic reprogramming of soldiers with obesity."
I have reviewed your revisions and I am pleased to acknowledge that all the comments and suggestions provided have been thoroughly addressed. Your efforts to rectify the concerns and improve the quality of the manuscript are commendable.

Experimental design

My comments and suggestions have been properly addressed. I have no further questions.

Validity of the findings

My comments have been properly addressed. I have no further questions.